# Transcription Factors and ncRNAs Associated with *CYP3A* Expression in Human Liver and Small Intestine Assessed with Weighted Gene Co-Expression Network Analysis

**DOI:** 10.3390/biomedicines10123061

**Published:** 2022-11-28

**Authors:** Huina Huang, Siqi Zhang, Xiaozhen Wen, Wolfgang Sadee, Danxin Wang, Siyao Yang, Liang Li

**Affiliations:** 1Department of Medical Genetics, School of Basic Medical Sciences, Southern Medical University, Guangzhou 510515, China; 2Center for Pharmacogenomics, Department of Cancer Biology and Genetics, College of Medicine, The Ohio State University, Columbus, OH 43210, USA; 3Center for Pharmacogenomics, Department of Pharmacotherapy and Translational Research, College of Pharmacy, University of Florida, Gainesville, FL 32610, USA; 4Experimental Education and Administration Center, School of Basic Medical Sciences, Southern Medical University, Guangzhou 510515, China

**Keywords:** cytochrome P450, WGCNA, transcription factor, long non-coding RNA, regulatory network

## Abstract

CYP3A4, CYP3A5, and CYP3A7, which are located in a multigene locus (*CYP3A*), play crucial roles in drug metabolism. To understand the highly variable hepatic expression of CYP3As, regulatory network analyses have focused on transcription factors (TFs). Since long non-coding RNAs (lncRNAs) likely contribute to such networks, we assessed the regulatory effects of both TFs and lncRNAs on *CYP3A* expression in the human liver and small intestine, main organs of CYP3A expression. Using weighted gene co-expression network analysis (WGCNA) of GTEx v8 RNA expression data and multiple stepwise regression analysis, we constructed TF-lncRNA-*CYP3A* co-expression networks. Multiple lncRNAs and TFs displayed robust associations with *CYP3A* expression that differed between liver and small intestines (LINC02499, HNF4A-AS1, AC027682.6, LOC102724153, and RP11-503C24.6), indicating that lncRNAs contribute to variance in *CYP3A* expression in both organs. Of these, HNF4A-AS1 had been experimentally demonstrated to affect *CYP3A* expression. Incorporating ncRNAs into *CYP3A* expression regulatory network revealed additional candidate TFs associated with *CYP3A* expression. These results serve as a guide for experimental studies on lncRNA-TF regulation of *CYP3A* expression in the liver and small intestines.

## 1. Introduction

As a key subfamily of cytochrome P450 (CYP) enzymes, CYP3A enzymes contribute to the metabolism of more than 50% of all marketed drugs [1]. Members of this subfamily include CYP3A4, CYP3A5, CYP3A7, and CYP3A43. CYP3A4 and CYP3A5 are the primary metabolizing enzymes of the CYP3A subfamily. CYP3A7 is primarily expressed in the fetus and newborn, but also exhibits relatively high expression in some adult livers [2]. The liver and small intestine are the main organs of drug metabolism and express high *CYP3A* levels, except for CYP3A43 [3,4], which is therefore not considered further here.

*CYP3A* expression displays substantial inter-individual variability in the human liver and small intestine [3,4], contributing to variable drug responses. Regulatory mechanisms at transcriptional and post-transcriptional levels appear to play the primary role in variable *CYP3A* expression [5]. It is critical to uncover the regulatory mechanism of *CYP3A* expression to improve the therapeutic effect and reduce the side effects of *CYP3A*-related drugs. By binding to *cis*-acting elements of *CYP3A*s, multiple transcription factors (TFs) and their networks can regulate *CYP3A* expression, including progesterone X receptor (PXR; NR1I2), constitutive androgen receptor (CAR; NR1I3), hepatocyte nuclear factor 4 alpha (HNF4α; NR2A1), vitamin D receptor (VDR; NR1I1), glucocorticoid receptor (GR; NR3C1), Yin Yang 1 (YY1), and estrogen receptor α (ESR1) [3,6,7]. Evidence suggests that the *CYP3A* gene cluster represents a regulome that is characterized by interacting enhancer/suppressor domains. Collins et al. [8] discovered that some distal *cis*-acting regulatory elements can simultaneously affect *CYP3A4*, *3A5*, and *3A7* expression because of chromatin looping. Therefore, combined analysis of *CYP3A*s can identify overlapping regulators, addressing coordinate expression.

Published network studies on *CYP3A* expression analyzed only the associations between TFs and *CYP3A* expression in liver [5,7,9]. Whereas the pervasive role of non-coding RNAs (ncRNAs) is well established [10,11], any regulatory roles of ncRNAs in *CYP3A* regulation remain unexplored. Therefore, inclusion of ncRNAs in co-expression networks of TFs and ncRNAs with *CYP3A*s has the potential to identify novel factors regulating *CYP3A* expression. Since *CYP3A* regulation may differ between the liver and small intestine [12], we included both tissues in the analysis to assess tissue-specific *CYP3A* regulation. NcRNAs can be divided into housekeeping ncRNAs and regulatory ncRNAs. Regulatory ncRNAs could be further classified as small non-coding RNAs (sncRNAs) (<200 nucleotides) and long non-coding RNAs (lncRNAs) (>200 nucleotides). The main classes of small ncRNAs are microRNAs (miRNAs), small interfering RNAs (siRNAs), and piwi-interacting RNAs (piRNAs). The lncRNAs, including cytoplasmic and nuclear lncRNAs, microRNA precursors, circular RNAs, and more, have diverse functions [13]. We focus on lncRNAs available in large databases generated with next generation RNAseq methodology.

Unlike methods that focus on single genes or a few genes [14], weighted gene co-expression network analysis (WGCNA) can transform gene expression data into co-expression modules, providing insight into signaling and regulatory networks [15]. In addition, WGCNA can identify RNAs with low abundance that may play important regulatory roles in biological responses [16]. Several studies have shown that WGCNA identifies modules and pathways and lncRNAs using gene clustering [9,17,18,19,20]. The Genotype-Tissue Expression (GTEx) database provides gene expression data across multiple human tissues [21]. After extracting GTEx v8 data of TFs and lncRNA expression in the liver and small intestine, we tested whether lncRNAs contribute to regulation of *CYP3A* expression, expanding previous CYP3A-TF networks with lncRNAs acting as potential network links and modulators.

## 2. Materials and Methods

### 2.1. GTEx v8 Data and Data Pre-Process

The GTEx v8 data dataset (https://gtexportal.org/home/ (accessed on 19 February 2020)) provides 56,200 genes for extraction from 193 liver and 175 small intestine samples [21]. Donor phenotype, RNAseq data (TPM, transcripts per million), and data processing information of samples were also obtained from the GTEx v8 dataset. Since most genes express multiple distinct transcripts, the sum of all transcripts TPM value of each gene was used as the expression value of the gene, and only the samples with RNA integrity number (RIN) ≥ 6 were selected. For each gene, we calculated the mean expression values for the liver and small intestine, and set the lower quartile of the means of all genes as the threshold. Next, genes with an expression value over the threshold (lower quartile of the mean) in more than 80% of the samples were selected, identifying 22,304 genes in the liver and 25,464 genes in the small intestines. Following the GTEx data processing method reported by Somekh et al. [22], we performed quantile normalization within each tissue, then transformed gene TPM to log2(TPM + 1), and adjusted for the effect of ischemic time by regressing the log2(TPM + 1) expression values with ischemic time. The annotations of gene types were obtained from Gencode v33 (https://www.gencodegenes.org/ (accessed on 14 March 2020)) [23]. Human transcription factors were downloaded from the animalTFDB 3.0 database (http://bioinfo.life.hust.edu.cn/HumanTFDB#!/ (accessed on 20 April 2020)) [24]. Genes encoding the RNAs were then divided into four groups: group 1 consisted of 22,304 genes in the liver; group 2 consisted of 25,464 genes in the small intestine; ncRNAs were then removed from group 1 to yield group 3 (19,273 genes) in the liver, and from group 2, generating group 4 (21,386 genes) in the small intestine. The demographics and gene types of the liver and small intestine samples in GTEx v8 dataset are shown in Table 1. The experimental approach is shown in Figure 1.

### 2.2. Construction of WGCNA Network and Detection of Modules

The construction of the WGCNA network and detection of modules were conducted using the WGCNA R package [15]. The power known as soft threshold (β) for the four groups of datasets was pre-calculated using the pickSoftThreshold function. BlockwiseModules, an automatic block-wise network construction function for large datasets, was used to construct an unsigned co-expression network and detect modules. The correlation method used in WGCNA was biweight midcorrelation (bicor), as it is more robust than the Pearson correlation. Modules, termed target modules, in each dataset contained three target genes (*CYP3A4*, *3A5*, and *3A7*), and these were extracted for further analysis.

### 2.3. Function Enrichment Analysis in Target Modules

Gene Ontology (GO) and Kyoto Encyclopedia of Genes and Genomes (KEGG) pathway enrichment analyses, based on hypergeometric distribution for genes in each target module with *p* value <0.05 and *q* value <0.05, were performed in R using the ClusterProfiler [25] R package.

### 2.4. Identification of TFs and ncRNAs Co-Expressed with CYP3As in Target Modules

For each module, WGCNA served to calculate the module eigengene E (ME), which is the first principal component of a given module and represents the gene expression profile of the module. For each gene, WGCNA calculated the correlation between gene expression value and ME as module membership (k_ME_). The k_ME_ value represents the degree of membership of the gene to the module and is highly correlated with the connectivity of the gene. For a given module, the larger the |k_ME_| of the gene, the more likely it is to be a hub gene [15]. In addition, WGCNA was used to calculate the weight value between two genes using the biweight midcorrelation method (bicor(A,B)). All gene pairs with a weight value >0.05 were retained. Amongst the four groups, the top 80 regulators (TFs and ncRNAs) were determined by selecting the highest weight values for each *CYP3A*, and these were chosen for further analysis. Their co-expression networks were performed using Cytoscape v.3.7.2 [26], and the positive and negative correlations were calculated using the bicor function in the WGCNA R package.

### 2.5. Functional Annotation of TFs and ncRNAs Co-Expressed with CYP3As in Target Modules

The ClusterProfiler [25] R package was used to annotate the top 80 regulators (TFs and ncRNAs) co-expressed with *CYP3A*s along with the *CYP3A*s in each target module from the four groups using the GO and KEGG function, animalTFDB 3.0 [24], miRBase (https://www.mirbase.org/ (accessed on 22 May 2020)) [27], and RNAcentral databases (https://rnacentral.org/ (accessed on 20 June 2020)) [28]. These were also combined with auxiliary annotations.

### 2.6. Multiple Stepwise Regression Analysis

A multiple stepwise regression analysis was performed using the SPSS software package (ver. 20.0; SPSS Inc., Chicago, IL, USA). The TPM values of the top 80 regulators (TFs and ncRNAs) co-expressed with *CYP3A*s for each *CYP3A* were set as independent variables, and the TPM values of each *CYP3A* were set as dependent variables. A *p* value less than 0.05 was considered statistically significant.

The R scripts are available on GitHub: https://github.com/servicemanli/CYP3A_regulatory_network.git (accessed on 22 August 2022).

## 3. Results

### 3.1. Construction of the WGCNA Network and Expression Modules

In this study, we applied WGCNA to explore CYP3A-TF networks with lncRNAs acting as potential network links and modulators, to determine whether *CYP3A*s expression networks in the liver and small intestine are altered by the inclusion of ncRNAs. Thus, we used WGCNA to analyze and compare the following four groups of data extracted from GTEx v8: group 1 with all genes in the liver, group 2 with all genes in the small intestine, group 3 with protein-coding genes in the liver, and group 4 with protein-coding genes in the small intestine (ncRNAs were removed in group 3 and 4). Shown in Appendix A, the lowest soft-thresholding powers were 4, 7, 4, and 7 for the four groups, respectively, for which the scale-free topology fit index (scale-free R^2^) reached 0.8. The appropriate soft-thresholding values emphasize strong gene–gene correlations. The correlation matrix was subsequently transformed into an adjacency matrix. Each adjacency matrix was normalized using a topological overlap measure (TOM). A dissimilarity matrix based on TOM was used to identify gene modules with a dynamic tree-cutting algorithm. Totals of 141, 48, 128, and 46 co-expression modules were identified in the four groups, respectively (Figure 2). In group 1, all three *CYP3As* were in the turquoise module, whereas in group 3, *CYP3A5* and *3A7* were in the turquoise module and *CYP3A4* was in the blue module. The number of genes within *CYP3A*-related modules also changed significantly from group 1 to group 3. These analyses revealed that liver modules containing *CYP3A*s and genes related *CYP3A*s changed after removal of the ncRNAs. The modules containing the *CYP3A*s from group 1 to group 4 and the numbers of genes within the modules are listed in Table 2.

### 3.2. Identification of TFs and ncRNAs Co-Expressed with CYP3As in Target Modules

We next searched for ncRNAs and TFs in the same expression modules as *CYP3A*s. For each *CYP3A*, the TFs and ncRNAs with weight values > 0.05 in the same module are shown in Appendix A. For group 1 and group 2 (liver and intestine + ncRNAs), the number of ncRNAs associated with each *CYP3A* exceeded the number of TFs (Appendix A). We compared the weight values of TFs and ncRNAs using an independent-sample *t*-test in groups 1 and 2, respectively. In group 1, for each *CYP3A*, there was no significant difference in the weight values between TFs and ncRNAs (data not shown). In group 2, for *CYP3A4* and *3A5*, the weight values of TFs were significantly higher than those of ncRNAs (*CYP3A4*: 0.12 ± 0.062 vs. 0.10 ± 0.046, *p* = 0.003; *CYP3A5*: 0.17 ± 0.057 vs. 0.16 ± 0.052, *p* = 3.10 × 10^−4^). For *CYP3A7* in group 2, no significant difference was found in the weight values between TFs and ncRNAs (data not shown). These results show that multiple ncRNAs can reach weight values with *CYP3A*s matching those of TFs in the liver. In the small intestine, TFs are more closely associated with *CYP3A4* and *3A5* than ncRNAs.

We then ordered the potential regulators by weight and analyzed the associations between the top 80 potential regulators and each *CYP3A* expression from group 1 to 4 via biweight midcorrelation method. As shown in Appendix A, most of the potential regulators are significantly associated with *CYP3A* expression, even after Bonferroni correction. If there are multiple *CYP3As* in one module, the regulators often have similar associations between these *CYP3As*, whether positive or negative associations. Among the associations with *CYP3As*, some lncRNAs had lower *p* values compared to TFs, such as lncRNA LINC02499 with *CYP3A5* (*p* = 5 × 10^−23^) in group 1; and lncRNA MIR194-2HG (miRNA precursor) with *CYP3A5* (*p* = 1 × 10^−98^) and lncRNA EGFR-AS1 (anti-sense RNA) with *CYP3A7* (*p* = 9 × 10^−38^) in group 2. These results indicate that lncRNAs are significantly associated with *CYP3A*s in the liver and small intestine (Appendix A). In addition, we found that most lncRNAs are positively associated with *CYP3A* expression, including LINC02499, MIR194-2HG, and EGFR-AS1.

After removal of ncRNAs in the liver, the modules containing *CYP3A*s were changed, and *CYP3A*-related TFs were changed. In group 1, the previously reported *CYP3A*-related TFs such as NR1I2, NR1I3, and ESR1 were in the turquoise module (same module as *CYP3A4*, *3A5* and *3A7*) [7,29,30]. However, in group 3, NR1I2 and NR1I3 were in the turquoise module (same module as *CYP3A5* and *3A7*), whereas ESR1 was in blue module (same module as *CYP3A4*). In the small intestine, *CYP3A*-containing modules and *CYP3A*-related TFs were similar before and after the removal of ncRNAs. These results suggest that the removal of ncRNAs from the *CYP3A* alters inferred regulatory networks, especially in the liver.

### 3.3. Functional Enrichment Analysis in Target Modules

Next, we aimed to evaluate the functions of genes within the modules co-expressed with *CYP3A*s. Functional enrichment analysis was performed on the seven modules containing *CYP3A*s (turquoise module in group 1, turquoise and red modules in group 2, turquoise and blue modules in group 3, blue and green modules in group 4) by GO and KEGG analysis. Appendix A shows the most significantly enriched category for each module. The gene-enriched biological processes (BPs) of these modules were predominantly biometabolism and biosynthesis. The gene-enriched molecular functions (MFs) of this module were mainly enzyme activity, binding with other molecules, and substance-transporting activities. The module-enriched cellular components (CCs) included mitochondrial matrix and brush border membrane. The seven modules were enriched with multiple functional gene entries related to the CYP450 family (Appendix A). These functional enrichment results of module genes support the reliability of the WGCNA-constructed network and the identified modules.

### 3.4. Comparison of CYP3A Associated TFs in the Liver and Small Intestine with and without ncRNAs

We evaluated whether TFs and *CYP3A* regulatory networks were altered in the absence of ncRNAs. For each *CYP3A*, we compared the numbers of TFs associated with *CYP3A* with and without ncRNAs in the liver and small intestine.

Focusing on liver expression with and without ncRNAs, most of the TFs associated with *CYP3As* overlapped with TFs found in group 1 when including ncRNAs (Appendix A and Figure 3). Group 1 also had a subset of TFs that were not in group 3. Compared to the overlapping TFs, some of these TFs had higher weight values with *CYP3As*, such as ZGPAT (weight value with *CYP3A4* was 0.16), ZNF385B (weight value with *CYP3A5* was 0.14), and ARID3C (weight value with *CYP3A5* was 0.14). We found that after removing ncRNAs, the numbers of TFs related to *CYP3A4*, *3A5*, and *3A7* in the liver decreased. These results indicate that omission of ncRNAs in the liver can lead to missed relationships TFs associated with *CYP3A* expression. In addition, the regulatory networks for the three *CYP3As* in the liver differed between group 1 with ncRNAs and group 3 without ncRNAs (Figure 4a,b).

In the small intestine, group 4 without ncRNAs, most TFs associated with *CYP3A4* and *3A5*, overlapped with TFs in group 3, including ncRNAs (Appendix A and Figure 3). Similarly, the regulatory networks for *CYP3A4* and *3A5* did not change substantially in small intestine upon removal of ncRNAs (Figure 4c,d).

### 3.5. Comparison of CYP3A-Associated TFs and ncRNAs between Three CYP3As in the Liver and Small Intestine

We next asked whether TFs and ncRNAs are shared between different *CYP3A*s in the liver and small intestine. In group 1 and group 2 (liver and intestines + ncRNAs), the numbers of TFs and ncRNAs associated with each *CYP3A* were compared among the three *CYP3As*. In group 1, most TFs and ncRNAs overlapped between all three *CYP3As* (Appendix A and Figure 5). In the small intestine, most TFs and ncRNAs overlapped only between *CYP3A4* and *3A5* (Appendix A and Figure 5). Since *CYP3A7* resides in a different module, its regulatory mode appears to differ from those of *CYP3A4* and *3A5*. These results indicate that *CYP3A4*, *3A5*, and *3A7* are under similar regulatory control in the liver. Meanwhile, *CYP3A4* and *3A5* but not *3A7* are under similar regulatory control in the small intestine.

### 3.6. Comparison of CYP3A Expression Network between Liver and Small Intestines

To compare *CYP3A*-associated TFs and ncRNAs in the liver and small intestine, TFs and ncRNAs associated with each *CYP3A* with a weight value greater than 0.05 from groups 1 and 2 were selected. As shown in Appendix A and Figure 6, most TFs and ncRNAs differ between the liver and small intestines for each *CYP3A*. *CYP3A4* and *3A5* are associated with more TFs and ncRNAs in the small intestines than in the liver. In addition, the average weight values of TFs and ncRNAs associated with *CYP3A*s in the small intestine were higher than that in liver. These results indicate that different TFs and ncRNAs are involved in the regulatory modes of *CYP3A* expression in the liver and small intestine.

### 3.7. Multiple Stepwise Regression Analysis for the Association with CYP3A4,5,7 RNA Expression Values

We reasoned that ncRNAs and TFs could explain part of the variance in *CYP3A*s. To this end, for the liver and small intestine, the top 80 regulators (TFs and ncRNAs) of each *CYP3A* (Appendix A) were selected. Of the 80 regulators, all the ncRNAs were lncRNAs. The TPM values (level of expression) of the 80 regulators were set as independent variables, and TPM values of each *CYP3A* were set as dependent variables. Then, we conducted a multiple stepwise regression analysis to evaluate the contribution of regulators to *CYP3A*’s expression variance. In the liver, the TFs and lncRNAs in the final models explained 68.1%, 69.5%, and 44.5% of the expression variance in *CYP3A4*, *3A5*, and *3A7*, respectively (Table 3). The results reveal that the top significant variables, ESR1, LINC02499, and HNF4A-AS1 explained 36.4%, 48.5%, and 23.8% of the expression variance in *CYP3A4*, *CYP3A5*, and *CYP3A7*, respectively (Table 3). After the removal of the lncRNAs from the top 80 variables, the TFs in the final models explained 45.1%, 55.3%, and 26.0% of the expression variance in *CYP3A4*, *3A5*, and *3A7*, respectively. The most significant TFs, ESR1, NR1I2, and ZBTB47, explained 36.4%, 40.5%, and 21.5% of expression variance in *CYP3A4*, *CYP3A5*, and *CYP3A7*, respectively (Appendix A).

In small intestine, TFs plus lncRNAs in the final models accounted for 95.7%, 98.1%, and 80.4% of the expression variance in *CYP3A4*, *3A5*, and *3A7*, respectively (Table 3). The results reveal that the significant variables LOC102724153 and RP11-503C24.6 explained 83.6% and 62.5% of the expression variance of *CYP3A4* and *CYP3A7*, respectively (Table 3). For *CYP3A5*, NR1I2 was the most significant variable (*p* = 2.7 × 10^−22^) associated with *CYP3A5* expression, and no other TFs or lncRNAs significantly accounted for the variance in *CYP3A5* expression. After the removal of the lncRNAs from top 80 variables, the TFs in the final models explained 93.8%, 96.0%, and 69.3% of the expression variance in *CYP3A4*, *3A5*, and *3A7*, respectively. The significant TFs CREB3L3, NR1I2, and NR6A1 explained 83.3%, 92.7%, and 56.8% of expression variance in *CYP3A4*, *CYP3A5*, and *CYP3A7*, respectively (Appendix A).

For each *CYP3A*, the weight values between the two lncRNAs accounting for the highest proportion of *CYP3A* expression variance and the TFs within the module are listed in Appendix A.

We conclude that, after the removal of lncRNAs, the proportions of regulators explaining expression variance in *CYP3A4*, *3A5*, and *3A7* in liver decreased by 23.0%, 14.2%, and 20.7%; and the decreases in regulators explaining expression variance in *CYP3A4*, *3A5*, and *3A7* in the small intestine were only 1.9%, 2.1%, and 11.1%, respectively. The results indicate that lncRNAs had a greater regulatory effect on *CYP3A*s in the liver than in the small intestine.

## 4. Discussion

In this study, we conducted a weighted gene co-expression network analysis to identify TFs and ncRNAs associated with *CYP3A*s in the liver and small intestine based on GTEx v8 data. The expression data were analyzed with and without ncRNAs in both the liver and small intestine. Bioinformatic databases were applied to further prioritize TFs and ncRNAs using multiple stepwise regression analysis to identify TFs and ncRNAs contributing to *CYP3A* expression in the liver and small intestine. Networks obtained with CYP3As and TFs alone are similar to those published earlier, and a series of previously reported TFs related to CYP3A, such as ESR1, HNF4α, and NR1I2, were also identified here [5,7,9]. Including ncRNAs with the analysis yielded an expanded set of TFs and multiple significant lncRNAs affecting *CYP3A* expression.

Our results show that multiple ncRNAs have weight values with *CYP3A*s similar to those of TFs in both liver and small intestine (Appendix A). Co-expression networks show that a series of lncRNAs and TFs are significantly associated with *CYP3A* expression even after Bonferroni correction in both tissues; some lncRNAs outranked TFs as potential regulatory factors (Appendix A). In the liver, the lncRNAs HNF4A-AS1, LINC02499, and RP11-669E14.4 showed the most significant associations with *CYP3A4*, *3A5*, and *3A7*, respectively (Appendix A). In the small intestine, LOC102724153, NR1I2 (a TF), and EGFR-AS1 ranked highest for *CYP3A4*, *3A5*, and *3A7*, respectively (Appendix A). Of these six top regulators, HNF4A-AS1 and NR1I2 were reported to be associated with *CYP3A* expression [29,31]. Experimental discovery of the role of lncRNA HNF4A-AS1 provides strong validation for the lncRNAs inferred by WGCNA [32,33,34]. These results reveal that multiple lncRNAs have robust associations with *CYP3A*s similar to that of TFs in both liver and small intestine. Most of the significant lncRNAs were positively associated with *CYP3A* expression.

LncRNAs can recruit regulatory protein complexes to a gene to regulate its transcription [10]. In addition, the lncRNA transcripts can bind to and regulate proteins [35]. Comparing regulatory networks of *CYP3A* expression with and without ncRNAs showed significant changes in hepatic *CYP3A* networks. Inclusion of ncRNAs (Group 1) identified more TFs than present in networks without ncRNAs (Group 3) (Figure 3 and Figure 4, Appendix A). Most ncRNAs in the same module as *CYP3A*s were lncRNAs. This result supports the hypothesis that lncRNAs indeed function as co-regulators in the liver.

A number of TFs regulating *CYP3A* expression had been identified, such as NR1I2, ESR1 and PPARA [5,7,9]. While these TFs were also identified here, our results indicate that adding lncRNAs to hepatic *CYP3A* regulatory networks not only reveals a role for lncRNAs but also identifies additional TFs candidates affecting *CYP3A* expression. In the small intestine, on the other hand, significant TFs in the *CYP3A4* and *3A5* networks were similar with and without ncRNAs in the analysis (Figure 3 and Figure 4, Appendix A). In addition, the mean weight values of TFs for *CYP3A4* and *3A5* were significantly higher than those of ncRNAs in the small intestine. These results suggest that ncRNAs may play a more prevalent role in the regulation of *CYP3A4* and *3A5* expression in the liver than in the small intestine.

The multigene locus with *CYP3A4*, *3A5*, and *3A7* contains overlapping promoter and enhancer regions as a result of chromatin looping [8]. Our results indicate that the three *CYP3A*s share most significant TFs and ncRNAs in the liver with similar weight values (Figure 5). These results indicate that *CYP3A4*, *3A5*, and *3A7* may have similar, likely overlapping regulatory modes. Collins et al. [8] and Wang et al. [7] reported enhancer sites associated with expression of more than one *CYP3A* gene and TFs affecting more than one *CYP3A*. On the other hand, a single regulator can have different targets for each individual *CYP3A*, resulting in co-expression with multiple *CYP3A*s. In the small intestine, *CYP3A4* and *3A5* shared most TFs and ncRNAs. For each regulator, the weight value with *CYP3A5* was significantly higher than the weight value with *CYP3A4* (Figure 5). This result suggests that the regulatory effects of TFs and ncRNAs on *CYP3A5* were stronger than that of *CYP3A4* in the small intestine. Considering that *CYP3A7* is in different modules from *CYP3A4* and *3A5*, the TFs and ncRNAs associated with *CYP3A7* differ from those of *CYP3A4* and *3A5*, suggesting different regulatory modes of *CYP3A4*, *3A5*, and *3A7* in the small intestine. In addition, each *CYP3A* has indifferent and more significantly associated TFs and ncRNAs in the small intestine than the liver (Figure 6), demonstrating distinct co-expression patterns of *CYP3A*s in the liver and small intestine.

We also performed multiple stepwise regression analysis to evaluate the contribution of each regulator to *CYP3A* expression variance. ESR1 was the most significant regulator affecting *CYP3A4* expression in the liver (Table 3). In prior studies, Wang et al. [7] found that ESR1 is a master regulator in the hepatic expression of *CYP3A4*, and our results confirm this finding. LINC02499 and HNF4A-AS1 are the top-ranking lncRNAs affecting *CYP3A5* and *3A7* expression, respectively (Table 3). Ma et al. [36] reported that LINC02499 expression was remarkably decreased in hepatocellular carcinoma (HCC) tissues compared to adjacent non-tumor tissues, and decreased LINC02499 was also significantly associated with poorer overall survival in the HCC cohort. The mechanisms by which LINC02499 regulates *CYP3A5* expression in the liver still require further investigation. Chen et al. [31] reported that knockdown of HNF4A-AS1 increases mRNA expression of basal levels of P450s, including *CYP3A4*, in HepaRG cells. These results confirmed that HNF4A-AS1 participates in the regulatory network of *CYP3A*s in the liver. In the small intestine, LOC102724153 and RP11-503C24.6 account for large proportions of the variance in *CYP3A4* and *3A7* expression, respectively (Table 3). In addition, after removal of ncRNAs, the proportion of *CYP3A* expression variance explained by regulatory factors decreased in the liver more than in the small intestine (Table 3, Appendix A). These results suggest that lncRNAs account for a portion of variance in *CYP3A* expression in the liver, and more markedly so in the small intestines (Table 3). We also analyzed the weight values between the lncRNAs and TFs in the same modules. Some lncRNA–TF weight values exceeded those between lncRNAs and *CYP3A*s (Appendix A), providing insight as to whether these lncRNAs might interact directly with *CYP3A*s or indirectly through TFs.

Predicting *CYP3A* expression as a guide for clinical drug therapy more broadly must consider diverse factors, including genetic polymorphisms, induction by xenobiotics, and clinical factors [37,38]. For example, *CYP3A5*3*, the nonfunctional allele of *CYP3A5*, can account for a large portion of variance in *CYP3A5* expression and pharmacokinetic variability of CYP3A5 substrates [39,40]. According to our results, lncRNAs should also be considered as important factors in *CYP3A* expression. By including lncRNAs, our study provides new insights into the *CYP3A* regulation that underlies the interindividual pharmacokinetic and pharmacodynamic variability of *CYP3A*s. Well-defined regulatory networks may enhance clinical predictions of CYP3A related drug metabolism, with lncRNAs as additional biomarkers of toxicity or metabolism [34,41].

GTEx v8 provides extensive expression data of protein-coding genes and ncRNA genes in multiple tissues, enabling us to identify TFs and lncRNAs affecting *CYP3A* expression in the liver and small intestine. Our results revealed that several lncRNAs have robust associations with *CYP3A*s expression in the liver and small intestine, and lncRNAs may play crucial roles in the *CYP3A* regulatory network in the small intestine, and even more so in the liver. Our study expanded on potential factors influencing *CYP3A* expression and highlighted a role of lncRNAs in contributing to tissue-specific *CYP3A* expression. Further experiments should be applied to verify the mechanism of lncRNAs significantly associated with *CYP3A* expression in regulating *CYP3A* expression.

Our study has several limitations. First, a series of miRNAs have been reported to affect *CYP3A* expression [42,43,44,45,46]. GTEx v8 provides only microRNA precursors but not mature miRNAs that require distinct assay methods and therefore could not be included in the network analysis. In our study, only three miRNA precursors were found to be co-expressed with *CYP3A* (Appendix A). Second, in multiple stepwise regression analysis, multi-collinearity existed among some independent variables, and the final model may have missed relevant independent variables. Last, all results were obtained using bioinformatic methods, and further experiments should be carried out to validate the results.

In summary, we used GTEx v8 datasets and WGCNA to generate a comprehensive catalog of regulator-associated *CYP3A* gene expression in the liver and small intestine.

## Figures and Tables

**Figure 1 biomedicines-10-03061-f001:**
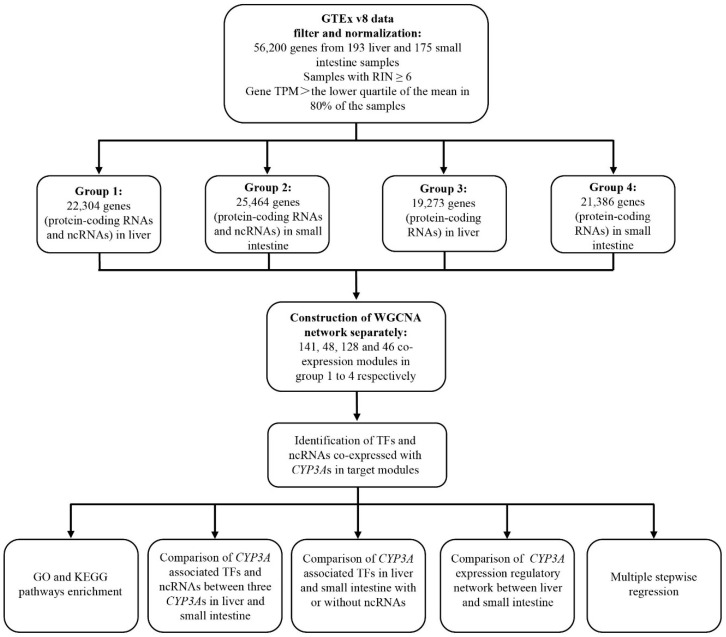
Study design. GTEx: Genotype-Tissue Expression project. RIN: RNA integrity number. TPM: transcripts per million. WGCNA: weighted gene co-expression network analysis. TFs: transcription factors. NcRNAs: non-coding RNAs.

**Figure 2 biomedicines-10-03061-f002:**
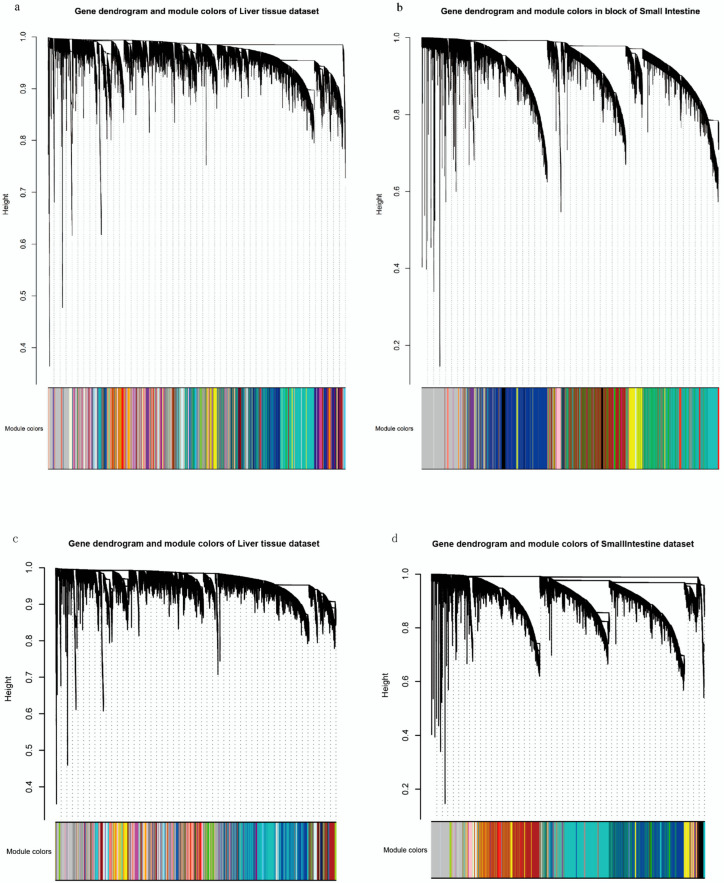
Gene clustering tree and module identification. The top halves of the four plots represent the gene clustering tree constructed based on topological overlap, and the height on the y-axis represents distance between modules; the higher the height value, the less likely it is that the two modules are co-expressed. The lower half of each of the four plots represents the gene modules, and each colored row represents a color-coded module that contains a group of highly connected genes. (**a**) The first group, (**b**) second group, (**c**) third group and (**d**) fourth group.

**Figure 3 biomedicines-10-03061-f003:**
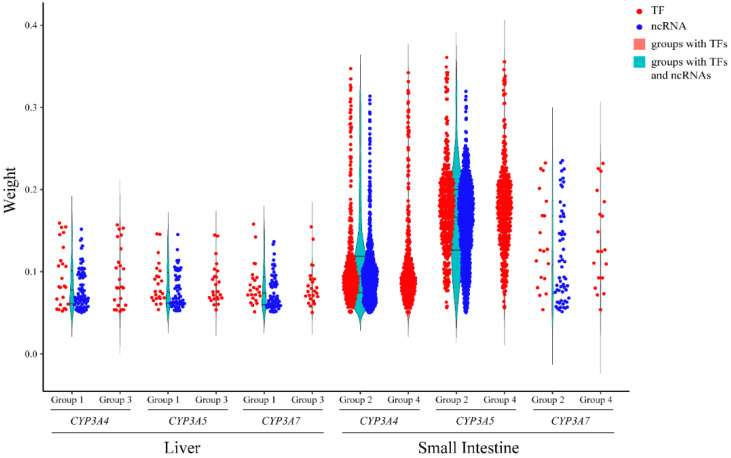
Comparison of *CYP3A*-associated transcription factors (TFs) in liver and small intestine with and without ncRNAs. The red dots represent the overlapping TFs associated with each *CYP3A* in liver and small intestine with and without ncRNAs. The blue dots represent the ncRNAs in liver and small intestine (group 1 and group 2). The red violin plots represent all TFs associated with each *CYP3A* in liver and small intestine (group 3 and group 4), and the green violin plots represent all TFs and ncRNAs associated with each *CYP3A* in liver and small intestine (group 1 and group 2). The y-axis has the weight values calculated by WGCNA.

**Figure 4 biomedicines-10-03061-f004:**
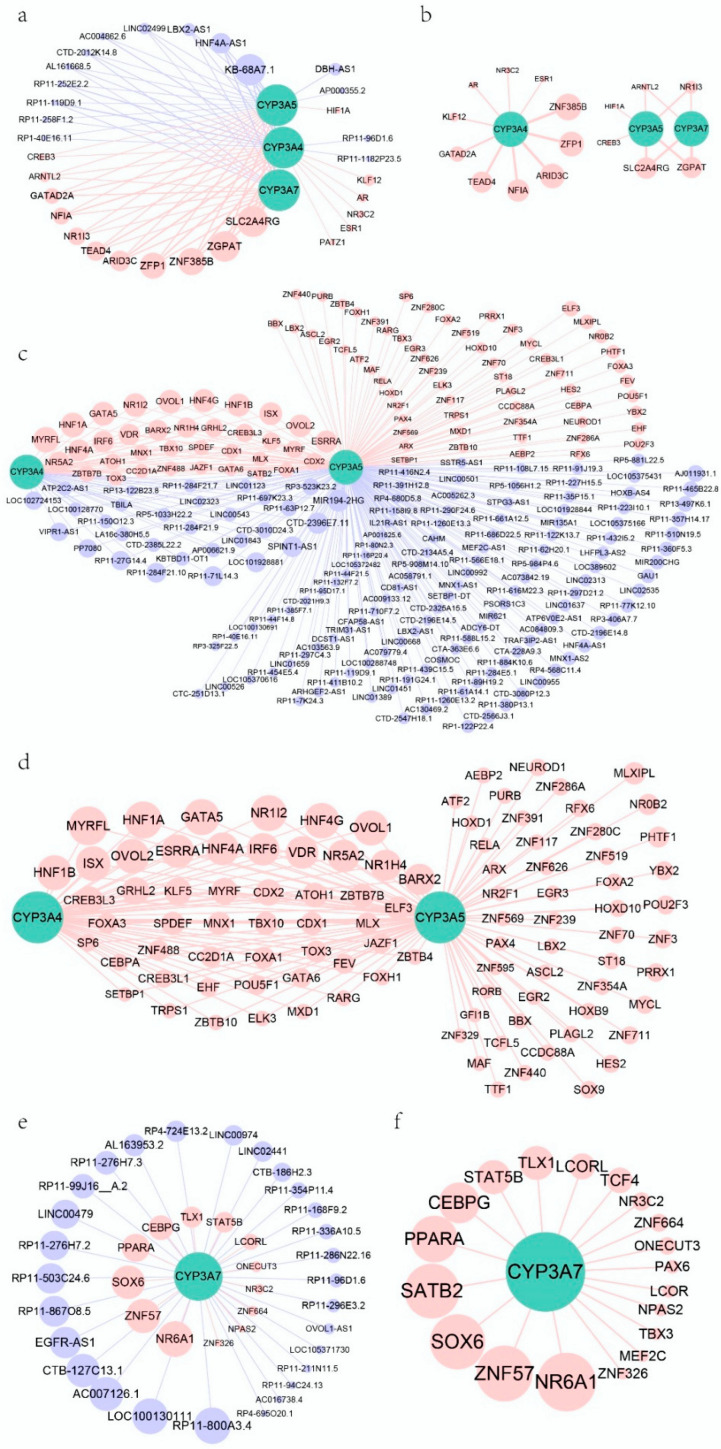
The direct co-expression relationships among TFs, lncRNAs, and *CYP3A4, 3A5,* and *3A7*. (**a**) The co-expression network of TFs, ncRNAs, and *CYP3A*s in group 1 and all TFs and ncRNAs with weight values > 0.1. (**b**) The co-expression network in group 3 (weight value > 0.1). (**c**) The co-expression network of TFs, ncRNAs, *CYP3A4*, and *3A5* in group 2. (**d**) The co-expression network of TFs, *CYP3A4*, and *3A5* in group 4 (weight value > 0.2). (**e**) The co-expression network of TFs, ncRNAs, and *CYP3A7* in group 2. (**f**) The co-expression network of TFs and *CYP3A7* in group 4 (weight value > 0.1). The thicker the edges in (**a**,**b**), the larger the weight values. The larger the dots in (**a**–**f**), the larger the weight values. The size of dots represents weight value with *CYP3A5* when there is more than one relationship among TFs, ncRNAs, and *CYP3A*s. The red and blue dots represent TFs and ncRNAs, respectively.

**Figure 5 biomedicines-10-03061-f005:**
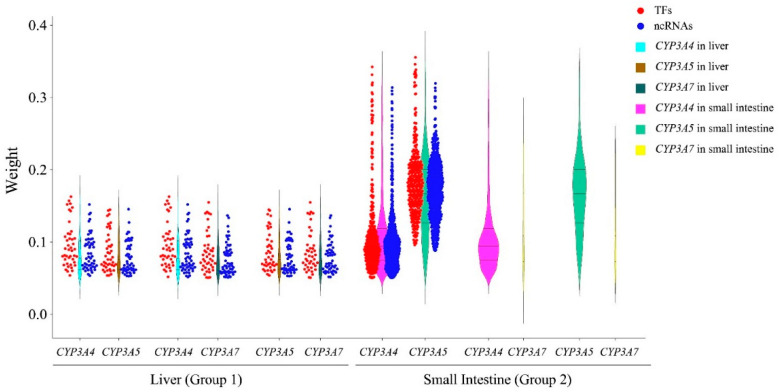
Comparison of *CYP3A*-associated TFs and ncRNAs with three *CYP3As* in the liver and small intestine. The y-axis is for the weight values calculated by WGCNA. Red and blue dots, respectively, represent the overlapping TFs and ncRNAs associated with two different *CYP3As* in the liver or small intestine. The violin plots with same color represent the same *CYP3A* in the same tissue.

**Figure 6 biomedicines-10-03061-f006:**
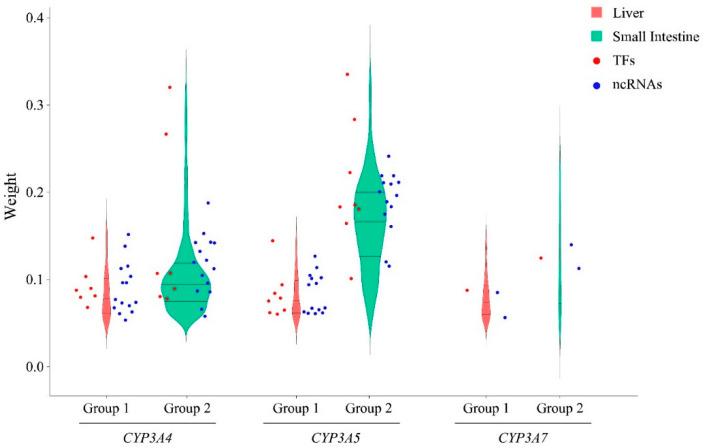
Comparison of *CYP3A* expression regulatory networks between liver and small intestine. The y-axis displays the weight values calculated by WGCNA. Red and blue dots represent the overlapping TFs and ncRNAs associated with each *CYP3A* in the liver and small intestine, respectively. The red and green violin plots represent all TFs and ncRNAs associated with each *CYP3A* in the liver and small intestine, respectively.

**Table 1 biomedicines-10-03061-t001:** Demographics and gene types of the liver and small intestine samples in GTEx v8 dataset.

	Liver (Mean ± SD)	Small Intestine (Mean ± SD)
Total number (*n*)	193	175
Age (years)	54.48 ± 11.04	47.66 ± 13.63
Sex (male/female) (*n*)	131/62	109/66
Race(White/Black/Asian/unknown)	169/19/4/1	144/27/2/2
Height (in)	68.11 ± 3.74	67.57 ± 3.80
Weight (lb)	179.03 ± 36.15	179.62 ± 34.47
BMI (kg/m^2^)	26.97 ± 4.14	27.51 ± 3.98
Liver diseases (Yes/No)	3/190	3/172
Gastrointestinal diseases (Yes/No)	0/193	0/175
Total number of genes (*n*)	56,200	
Protein-coding genes (*n*) ^a^	19,646	
Pseudogenes (*n*) ^a^	14,897	
To be Experimentally Confirmed (TEC) ^a^	1008	
Unidentified genes (*n*) ^b^	497	
NcRNAs (*n*) ^a,c^	20,152	
Long non-coding RNAs (lncRNA) (*n*) ^a,d^	13,731	
MicroRNAs precursors (miRNA) (*n*) ^a,d^	1576	
Miscellaneous other RNAs (misc_RNA) (*n*) ^a,d^	2007	
Small nuclear RNAs (snRNA) (*n*) ^a,d^	1864	
Small nucleolar RNAs (snoRNA) (*n*) ^a,d^	847	
Ribosomal RNAs (rRNA) (*n*) a,d	51	
Small Cajal body-specific RNAs (scaRNA) (*n*) ^a,d^	40	
Mitochondrial transfer RNAs (Mt_tRNA) (*n*) ^a,d^	22	
Mitochondrial ribosomal RNAs (Mt_rRNA) (*n*) ^a,d^	2	
Ribozymes (*n*) ^a,d^	6	
Small non-coding RNAs (sRNA) (*n*) ^a,d^	4	
Small cytoplasmic RNA (scRNA) (*n*) ^a,d^	1	
VaultRNA (vtRNA) (*n*) ^a,d^	1	

^a^ The annotations of gene types were obtained from Gencode v33 (https://www.gencodegenes.org/ accessed on 7 July 2022). ^b^ Gene types cannot be obtained in Gencode v33 database. NcRNAs ^d^ are the further classification of ncRNAs. ^c^ GTEx: Genotype-Tissue Expression project. NcRNAs: non-coding RNAs. BMI: Body mass index.

**Table 2 biomedicines-10-03061-t002:** Modules containing *CYP3As* identified from group 1 to 4 using weighted gene co-expression network analysis.

Group	Tissue	Gene Type	Module	Module Size	*CYP3A*	Number of TFs	Number of ncRNAs
Group 1	Liver	All genes	Turquoise	3103	*CYP3A4*, *3A5* and *3A7*	161	390 lncRNAs and 5 miRNAs
Group 2	Small intestine	All genes	Turquoise	4866	*CYP3A4* and *3A5*	339	658 lncRNAs and 3 miRNAs
			Red	1437	*CYP3A7*	55	257 lncRNAs and 4 miRNAs
Group 3	Liver	Protein coding genes	Turquoise	3090	*CYP3A5* and *3A7*	223	NA
			Blue	1991	*CYP3A4*	109	NA
Group 4	Small intestine	Protein coding genes	Blue	4078	*CYP3A4* and *3A5*	358	NA
			Green	1809	*CYP3A7*	83	NA

All genes in group 1 and 2 represent the genes that meet the criteria, including protein-coding genes and ncRNAs. The numbers for group 1 and 2 are 22,304 and 25,464, respectively. Protein-coding genes in group 3 and 4 represent the genes that meet the criteria, excluding ncRNAs. The numbers for group 3 and 4 are 19,273 and 21,386, respectively. NA represents not applicable. LncRNAs: Long non-coding RNAs. MiRNAs: microRNAs.

**Table 3 biomedicines-10-03061-t003:** Transcriptional factors and non-coding RNAs contributing to *CYP3A* expression in the liver and the small intestine.

Liver (*n* = 193)							
Dependent Variable	Independent Variable	Type	R-Square	*p*-Value	R-Square Change	Beta	*p*-Value
CYP3A4			68.11%	3.19 × 10^−36^			
	ESR1	humanTF			36.36%	0.488	1.44 × 10^−9^
	DBH-AS1	lncRNA			0.80%	−0.462	1.23 × 10^−6^
	AL161668.4	lncRNA			2.17%	−0.411	3.74 × 10^−6^
	AC008537.3	lncRNA			1.94%	0.334	3.88 × 10^−6^
	HNF4A-AS1	lncRNA			2.60%	0.515	5.18 × 10^−6^
	AC027682.6	lncRNA			2.48%	−0.346	1.29 × 10^−4^
	HIF1A	humanTF			0.86%	−0.262	1.84 × 10^−3^
	AC004160.2	lncRNA			3.95%	0.186	2.75 × 10^−3^
	LINC02499	lncRNA			2.59%	−0.266	6.23 × 10^−3^
	CTD-2325A15.5	lncRNA			1.46%	−0.172	1.12 × 10^−2^
	AC122713.2	lncRNA			0.94%	−0.179	1.13 × 10^−2^
	ZNF385B	humanTF			0.79%	0.234	1.82 × 10^−2^
	AL359715.3	lncRNA			0.84%	0.146	2.11 × 10^−2^
	HMGB3	humanTF			0.82%	−0.159	3.40 × 10^−2^
	ZGPAT	humanTF			0.73%	0.221	3.53 × 10^−2^
CYP3A5			69.52%	2.10 × 10^−39^			
	LINC02499	lncRNA			48.48%	0.514	3.84 × 10^−8^
	AL161668.4	lncRNA			1.50%	−0.407	9.04 × 10^−7^
	HLF	humanTF			2.48%	0.302	2.16 × 10^−4^
	CTD-2325A15.5	lncRNA			1.77%	−0.242	2.49 × 10^−4^
	AL122035.2	lncRNA			0.71%	0.211	9.84 × 10^−4^
	HMGB3	humanTF			1.19%	−0.214	2.67 × 10^−3^
	ZNF680	humanTF			0.86%	−0.203	3.44 × 10^−3^
	NFYC	humanTF			0.74%	0.174	8.33 × 10^−3^
	AC008537.3	lncRNA			0.99%	0.179	1.11 × 10^−2^
	NR1I2	humanTF			0.41%	0.187	1.86 × 10^−2^
	GCFC2	humanTF			2.01%	0.157	2.45 × 10^−2^
	ZNF473	humanTF			0.62%	0.146	2.48 × 10^−2^
	AC137056.1	lncRNA			0.47%	−0.142	3.26 × 10^−2^
CYP3A7			44.52%	1.64 × 10^−19^			
	HNF4A-AS1	lncRNA			23.84%	0.875	1.84 × 10^−8^
	DCXR-DT	lncRNA			4.53%	−0.447	3.24 × 10^−5^
	AL161668.4	lncRNA			2.32%	−0.431	8.09 × 10^−5^
	AL117382.3	lncRNA			1.52%	−0.267	1.83 × 10^−2^
	AL513327.1	lncRNA			1.98%	−0.169	2.49 × 10^−2^
	AC104809.1	lncRNA			1.48%	0.193	2.88 × 10^−2^
	AC083841.1	lncRNA			1.44%	0.186	3.04 × 10^−2^
	NR3C2	humanTF			2.48%	0.154	7.77 × 10^−2^
	RP11-669E14.4	lncRNA			4.94%	0.151	9.43 × 10^−2^
Small Intestine (*n* = 175)							
Dependent Variable	Independent Variable	Type	R-Square	*p*-Value	R-Square change	Beta	*p*-Value
CYP3A4			95.67%	3.18 × 10^−101^			
	NR1I2	humanTF			1.46%	0.937	1.05 × 10^−13^
	CREB3L3	humanTF			2.34%	0.427	1.28 × 10^−8^
	LOC102724153	lncRNA			83.59%	0.362	5.84 × 10^−7^
	FOXA3	humanTF			1.52%	−0.323	1.23 × 10^−5^
	BARX2	humanTF			0.91%	−0.293	2.24 × 10^−5^
	CTD-2547H18.1	lncRNA			0.63%	0.207	1.03 × 10^−4^
	GATA6	humanTF			0.19%	0.186	1.49 × 10^−4^
	LINC02323	lncRNA			3.60%	−0.198	1.95 × 10^−4^
	OVOL2	humanTF			0.21%	−0.298	3.63 × 10^−4^
	RP11-284F21.10	lncRNA			0.20%	−0.214	1.93 × 10^−3^
	SPINT1-AS1	lncRNA			0.17%	0.209	3.36 × 10^−3^
	LINC02313	lncRNA			0.28%	0.121	4.17 × 10^−3^
	LOC105375431	lncRNA			0.43%	−0.123	1.60 × 10^−2^
	MNX1-AS2	lncRNA			0.15%	−0.116	1.93 × 10^−2^
CYP3A5			98.09%	7.01 × 10^−127^			
	NR1I2	humanTF			0.64%	0.771	2.67 × 10^−22^
	RP11-63P12.7	lncRNA			1.64%	0.255	4.15 × 10^−11^
	MLXIPL	humanTF			0.55%	0.132	1.13 × 10^−9^
	HNF1B	humanTF			0.28%	0.409	4.42 × 10^−9^
	RP3-523K23.2	lncRNA			0.15%	−0.264	1.94 × 10^−6^
	ZBTB7B	humanTF			0.20%	−0.190	1.25 × 10^−5^
	KBTBD11-OT1	lncRNA			0.14%	−0.137	1.29 × 10^−5^
	RP11-150O12.3	lncRNA			0.25%	−0.143	1.07 × 10^−4^
	RP11-284F21.9	lncRNA			0.30%	0.127	7.77 × 10^−4^
	TBILA	lncRNA			0.12%	0.121	1.46 × 10^−3^
	TBX10	humanTF			0.38%	−0.107	3.69 × 10^−3^
	RP11-284F21.10	lncRNA			0.08%	−0.138	7.97 × 10^−3^
	LINC00543	lncRNA			0.35%	−0.093	8.63 × 10^−3^
	MNX1	humanTF			0.08%	0.122	1.33 × 10^−2^
	RP13-497K6.1	lncRNA			0.11%	0.050	1.52 × 10^−2^
	LOC100128770	lncRNA			0.05%	0.074	4.77 × 10^−2^
CYP3A7			80.38%	4.53 × 10^−50^			
	RP11-94C24.13	lncRNA			6.16%	0.470	5.77 × 10^−9^
	FOXD1	humanTF			2.04%	−0.347	3.14 × 10^−7^
	NR6A1	humanTF			0.91%	0.367	2.38 × 10^−4^
	NR3C2	humanTF			2.01%	0.307	2.40 × 10^−4^
	ONECUT3	humanTF			0.77%	−0.209	8.86 × 10^−4^
	RP11-503C24.6	lncRNA			62.50%	0.345	9.47 × 10^−4^
	RP11-794G24.1	lncRNA			1.78%	−0.234	1.50 × 10^−3^
	ZNF664	humanTF			0.82%	−0.225	1.94 × 10^−3^
	RP11-459I19.1	lncRNA			0.76%	−0.212	2.21 × 10^−3^
	EGFR-AS1	lncRNA			0.60%	0.310	5.12 × 10^−3^
	LOC105373958	lncRNA			0.65%	0.180	5.81 × 10^−3^
	RAG1	humanTF			0.69%	0.190	1.04 × 10^−2^
	RP11-211N11.5	lncRNA			0.70%	−0.164	1.78 × 10^−2^

Βeta: standardized coefficients.

## Data Availability

Publicly available datasets were analyzed in this study. This data can be found here: the GTEx v8 dataset (https://gtexportal.org/home/ (accessed on 19 February 2020)).

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
