# Peer review of "Transcription Factors and ncRNAs Associated with CYP3A Expression in Human Liver and Small Intestine Assessed with Weighted Gene Co-Expression Network Analysis"

_biomedicines, 2022, doi:10.3390/biomedicines10123061_

Round 1

Reviewer 1 Report

Huang et al identified, only by bioinformatics methods, transcription factors and lncRNAs associated with the expression of some cytochrome P450 genes in liver and small intestine.

Bioinformatic analyses have been correctly performed (for example GTEX data extraction and WGCNA), but the presentation of results resulted confused for a reader. I would suggest:

Major concerns:

1) In this version of the manuscript, the subject matter does not seem of considerable interest. Therefore, Authors should better present the clinical benefits and repercussions their findings have.

2) Furthermore, in each paragraph of the results, it should be better explained what has been achieved. In fact, there are many names, numbers, variables, data (scattered among main manuscript and supplementary materials) that confuse the reader. Introductory sentences should be added to the results to understand why those analyses were made and concluding sentences to summarize what was achieved.

3) It is not clear why the Prediction of the Intracellular Location of lncRNAs was performed. The paragraph 3.8 did not describe the results, but simply link to supplementary materials. Authors should explain why and discuss these results, otherwise this paragraph can be removed, as it adds nothing of interest.

Minor concerns:

4) In Introduction, please better describe WGCNA and its advantages, especially for lncRNA studies (and cite WGCNA-lncRNA studies, in addition to refs 15,16,17)

5) RNALocate, iLoc-lncRNA, Locate-R and other tool at paragraph 2.7 should be cited and described. A paragraph of only web links is not acceptable.

Author Response

Response to Reviewer 1 Comments

Point 1: In this version of the manuscript, the subject matter does not seem of considerable interest. Therefore, Authors should better present the clinical benefits and repercussions their findings have.

Response 1: We appreciate your comments. We have added comments on the clinical significance of our study in the second paragraph of introduction and seventh paragraph of discussion.

Point 2: Furthermore, in each paragraph of the results, it should be better explained what has been achieved. In fact, there are many names, numbers, variables, data (scattered among main manuscript and supplementary materials) that confuse the reader. Introductory sentences should be added to the results to understand why those analyses were made and concluding sentences to summarize what was achieved.

Response 2: For each paragraph of the results, we have added the introductory sentences at the beginning of each paragraph and concluding remarks at the end of each paragraph, improving the readability.

Point 3: It is not clear why the Prediction of the Intracellular Location of lncRNAs was performed. The paragraph 3.8 did not describe the results, but simply link to supplementary materials. Authors should explain why and discuss these results, otherwise this paragraph can be removed, as it adds nothing of interest.

Response 3: We have deleted these results.

Point 4: In Introduction, please better describe WGCNA and its advantages, especially for lncRNA studies (and cite WGCNA-lncRNA studies, in addition to refs 15,16,17)

Response 4: In the fourth paragraph of the introduction, we have described WGCNA and its advantages. Unlike other methods that can only analyze a single gene or a few genes, WGCNA transforms gene expression data into co-expression modules, providing insights into signaling networks that might be responsible for phenotypic traits of interest. WGCNA is a systems biological method, based on RNAseq data to discover the relationship between networks, genes, and phenotypes. Previous studies have shown that WGCNA offers modules and pathways relevant to disease WGCNA has proven suitable for analyzing lncRNA co-expression networks. We have addressed this briefly and cited WGCNA-lncRNA studies in the fourth paragraph of introduction section.

Point 5: RNALocate, iLoc-lncRNA, Locate-R and other tool at paragraph 2.7 should be cited and described. A paragraph of only web links is not acceptable.

Response 5: RNALocate, iLoc-lncRNA, Locate-R are deleted. For each other tool in the MATERIALS AND METHODS, we have cited the paper and given the weblink. Paragraph 2.7 Databases Websites has been deleted.

Reviewer 2 Report

In this manuscript, the authors perform bioinformatic analysis on the multigene locus CYP3A with respect to not only transcription factors but also lncRNAs. According to their analysis of liver and small intestine, incorporating ncRNAs into the proposed regulatory network of CYP3A reveals candidate transcription factors associated with CYP3A expression. The authors suggest these bioinformatic analyses can serve as a guide for experimental studies on these potential lncRNA-transcription factor regulation of CYP3A expression in liver and small intestines. 

This study is somewhat preliminary and largely descriptive. It will be of interest to a limited subset of specialists, who may want to follow-up with more concrete, experimental tests of certain proposed interactions. This study would be of greater interest if the authors could show, at least in principle, that their analysis did reveal, by experimental testing and validation, that at least one novel lncRNA-TF network component regulates CYP3A expression. 

Author Response

Response to Reviewer 2 Comments

Point 1: In this manuscript, the authors perform bioinformatic analysis on the multigene locus CYP3A with respect to not only transcription factors but also lncRNAs. According to their analysis of liver and small intestine, incorporating ncRNAs into the proposed regulatory network of CYP3A reveals candidate transcription factors associated with CYP3A expression. The authors suggest these bioinformatic analyses can serve as a guide for experimental studies on these potential lncRNA-transcription factor regulation of CYP3A expression in liver and small intestines.

This study is somewhat preliminary and largely descriptive. It will be of interest to a limited subset of specialists, who may want to follow-up with more concrete, experimental tests of certain proposed interactions. This study would be of greater interest if the authors could show, at least in principle, that their analysis did reveal, by experimental testing and validation, that at least one novel lncRNA-TF network component regulates CYP3A expression.

Response 1: Our study for the first time provides robust estimates of the involvement of lncRNAs by inference of co-expression networks on a large scale. Previous work has experimentally focused on individual lncRNAs, identifying HNF4A-AS1 as a regulator of CYP3A expression, including functional consequences on drug effects[reference 31,32,33]. This provide strong validation of our approach, identifying a number of lncRNAs showing strong associations with CYP3A expression. Experts in the field will be able to test these lncRNAs experimentally. We have started this work but expect results only within 1-2 years.

Reviewer 3 Report

It was a pleasure to review this article of importance in its field. I found the research question was clearly stated and well expressed objectives. The methods applied were seems to be appropriate and fully described in the article. Similarly, the results section well furnished, illustrative and self-explanatory enough, to answer the questions posed at the beginning. However, some minor important issues arise from the manuscript at its present state:

1) Figure 4 needs to be made better readable.

Author Response

Response to Reviewer 3 Comments

Point 1: It was a pleasure to review this article of importance in its field. I found the research question was clearly stated and well expressed objectives. The methods applied were seems to be appropriate and fully described in the article. Similarly, the results section well furnished, illustrative and self-explanatory enough, to answer the questions posed at the beginning. However, some minor important issues arise from the manuscript at its present state: 1) Figure 4 needs to be made better readable.

Response 1: We appreciate your comments. In the previous manuscript, too many TFs and lncRNAs (306 TFs and 544 lncRNAs) had been shown to be associated with CYP3A genes in the small intestine (Figure 4 c and d). In the revised manuscript, we restrict TFs and lncRNAs with weight value >0.2 (96 TFs and 152 lncRNAs) in Figure 4 c and d. In addition, we adjusted the color of dots and edges.

Round 2

Reviewer 1 Report

The authors have modified the manuscript in accordance with all my comments, so I suggest publication.

Reviewer 2 Report

Authors have adequately responded to concerns.